# Epigenetics Regulation in Responses to Abiotic Factors in Plant Species: A Systematic Review

**DOI:** 10.3390/plants13152082

**Published:** 2024-07-27

**Authors:** Geane Santos da Costa, Amanda Freitas Cerqueira, Carolina Reis de Brito, Marcelo Schramm Mielke, Fernanda Amato Gaiotto

**Affiliations:** Laboratório de Ecologia Aplicada à Conservação, Departamento de Ciências Biológicas, Universidade Estadual de Santa Cruz, Rodovia Jorge Amado km 16, Ilhéus, BA 45662-900, Brazil; gscosta@uesc.br (G.S.d.C.);

**Keywords:** epigenetic, abiotic stress, methylated DNA, histone modification, microRNAs

## Abstract

Plants have several mechanisms to adapt or acclimate to environmental stress. Morphological, physiological, or genetic changes are examples of complex plant responses. In recent years, our understanding of the role of epigenetic regulation, which encompasses changes that do not alter the DNA sequence, as an adaptive mechanism in response to stressful conditions has advanced significantly. Some studies elucidated and synthesized epigenetic mechanisms and their relationships with environmental change, while others explored the interplay between epigenetic modifications and environmental shifts, aiming to deepen our understanding of these complex processes. In this study, we performed a systematic review of the literature to analyze the progression of epigenetics studies on plant species’ responses to abiotic factors. We also aimed to identify the most studied species, the type of abiotic factor studied, and the epigenetic technique most used in the scientific literature. For this, a search for articles in databases was carried out, and after analyzing them using pre-established inclusion criteria, a total of 401 studies were found. The most studied species were *Arabidopsis thaliana* and *Oryza sativa*, highlighting the gap in studies of non-economic and tropical plant species. Methylome DNA sequencing is the main technique used for the detection of epigenetic interactions in published studies. Furthermore, most studies sought to understand the plant responses to abiotic changes in temperature, water, and salinity. It is worth emphasizing further research is necessary to establish a correlation between epigenetic responses and abiotic factors, such as extreme temperatures and light, associated with climate change.

## 1. Introduction

The term epigenetics was coined by researcher Waddington in 1947. The author defined it as the most appropriate word to define the branch of biology that studies the causality of interactions between genes and their products, which determines the organism’s phenotype [1]. Epigenetics today is better defined. After several adjustments between name and definition, it is clear now that epigenetics means changes in gene expression without changes in DNA sequences [2]. These changes are known to modulate gene expression in plants, many of which are responsible for development and phenotypic plasticity [3]. The most studied epigenetic mechanisms are DNA methylation, histone modifications, and non-coding RNAs (nRNAs).

DNA methylation is an epigenetic mechanism involved in processes such as transcriptional gene silencing, gene expression, recombination, and genome stability. It consists of an addition of a methyl group to the gene sequence. Methylation can be achieved by the addition of *5-methylcytosine* (*5-mC*), *N4-methylcytosine*, and *N6-methylcytosine*. In plants, *5-mC* is commonly added in sequences such as *CG*, *CHG*, and *CHH* (H = *A*, *T*, or *C*) [4,5,6,7]. This methylation is maintained by several methyltransferases, depending on the context [8,9,10].

Histones are known to package DNA, protect it, and provide access to its replication and transcription [11]. Modifications of these histones usually occur through acetylation and methylation. These modifications occur most frequently in histone 3 (*H3*) or histone 4 (*H4*), which are responsible for modifications in the transcription that activate or inhibit [12,13].

On the other hand, non-coding RNAs (nRNA) are transcripts that do not encode proteins. They are classified according to the size of their chain as long-stranded RNA (lncRNA) and short-stranded RNA. In the present review, we considered siRNA (small interfering RNA), miRNA (microRNA), and piRNA (piwi-interacting RNA). siRNAs are short double-stranded RNA molecules, typically 20–25 nucleotides long, that play a role in the RNA interference (RNAi) pathway. They are formed by the cleavage of longer double-stranded RNA molecules by an enzyme called Dicer. One of the strands is then loaded onto an RNA-induced silencing complex (RISC), which guides the siRNA to a complementary mRNA target. The RISC then catalyzes the cleavage of the mRNA, leading to its degradation and subsequent gene silencing. 

miRNAs are also small non-coding RNA molecules, approximately 21 nucleotides long, that are involved in post-transcriptional gene regulation. They are transcribed from longer RNA precursors called pri-miRNAs, which are processed in the nucleus to form pre-miRNAs. These pre-miRNAs are then exported to the cytoplasm, where they are further processed into mature miRNAs by Dicer. Mature miRNAs are incorporated into the RISC, where they typically bind to the 3′ untranslated region (UTR) of target mRNAs, leading to translational repression or mRNA degradation. 

piRNAs are a class of small non-coding RNA molecules, approximately 26–31 nucleotides long, that interact with a protein called PIWI (P-element induced wimpy testis in Drosophila). They are primarily found in the germ cells of animals and are thought to play a role in transposon silencing and genome stability. piRNAs are transcribed from genomic regions called piRNA clusters and are processed into mature piRNAs by a mechanism that is not fully understood. The piRNA–PIWI complex is thought to recognize and silence transposable elements by promoting heterochromatin formation or by cleaving target transcripts. Studies show that these nRNA are involved in gene expression control and epigenetics. Thus, they are involved in several regulatory functions in eukaryotes [14,15].

Plants have several mechanisms that enable their adaptation or acclimation to environmental changes. In this regard, stress caused by abiotic factors such as temperature, water, light, and salinity threaten plant species from both ecological and economic perspectives. Several research efforts are currently underway to unravel the mechanisms that determine genetic/epigenetic/phenotypic variation in response to these fluctuating abiotic stressors [16,17,18,19]. Many studies have attempted to understand plant responses to environmental change from ecological and economic perspectives [16], but there is a gap in studies presenting the molecular aspects. Advancing this knowledge will allow us to use it to develop plants tolerant to the aforementioned changes [18]. This is because epigenetic changes regulate gene expression and thus influence phenotypic plasticity, allowing an organism to adapt to changes in its environment [19].

Given the current concern about climate change, studies revealing epigenetic responses to abiotic factors are becoming increasingly significant [18,19]. This is because these results can help in understanding how plant responses can be controlled and consequently how they help in biodiversity conservation strategies. Given the increasing number of studies on this topic, it is important to conduct studies that analyze what has already been completed and what gaps remain to be filled. From this perspective, other reviews [20,21,22] have attempted to analyze epigenetic effects essentially in terms of biotic effects. However, there are still gaps to be filled when it comes to epigenetic mechanisms and abiotic effects in plants.

Therefore, the goal of this review was to examine the temporal and geographic aspects of epigenetic research related to abiotic stressors to evaluate the research efforts in different plant species. We aimed to identify which species are most studied and which abiotic factors and epigenetic techniques are the most used. Hence, we highlight commonalities and deficiencies in global research on this topic. By these means, we attempt to provide a scenario of the research to date and provide directions to develop appropriate research on the relationship between epigenetics and abiotic factors in plant species.

## 2. Results

### 2.1. Number of Articles and Species Studied

The initial search for studies on epigenetic regulation of abiotic factors in plant species revealed 5772 publications in the two used databases. Considering the exclusion and inclusion criteria, a total of 523 articles were selected, of which only 401 met the eligibility criteria to be included in the subsequent analyses.

A substantial number of species (*n* = 142) were studied in the selected articles, however, 88 of these were only cited in a single study, and 7 were part of studies involving multiple species (Table 1). Three species stood out in terms of the number of publications, with the model species *Arabidopsis thaliana* (*n* = 90) having the highest number of publications, followed by *Oryza sativa* (*n* = 38) and *Zea mays* (*n* = 26) (Table 1). Only one study each was found for species such as *Brassica juncea*, *Hibiscus cannabinus*, *Pinus sylvestris* and *Triticum durum*.

### 2.2. Epigenetic Technique and Abiotic Factors Best Studied

Most studies investigated epigenetic processes through DNA methylation (*n* = 207), while 84 studies investigated histone modifications and miRNAs. Additionally, our analysis revealed that 23 studies employed two of these techniques concurrently, whereas merely three studies utilized all three techniques within a single publication.

In relation to abiotic factors, most studies, 27% (*n* = 107), attempted to correlate species responses to temperature fluctuations, tracking that environmental factor, while 25% (*n* = 100) investigated the differences in species responses to water availability, 21% (*n* = 83) analyzed the influence of salinity, and only 7% (*n* = 28) studied the light factor. Moreover, 17% (*n* = 67) of the studies referred to two or more factors and 3% *(n* = 11) of the studies dealt with other abiotic factors such as gases and pollutants.

When we analyze the distribution of studies considering both techniques and abiotic factors simultaneously, the scenario is very similar to the one we found individually. But in studies using only DNA methylation, the number of articles concerning temperature *(n* = 53) surpassed those related to water *(n* = 50) and salinity (*n* = 45). For modified histones, we observed a higher concentration of studies analyzing the influence of water, salinity, and temperature. The articles that used the miRNA technique had more associations with the influence of water, temperature, and multiple stressors (Figure 1).

### 2.3. Time Trend of the Publications

The number of publications about epigenetics regulation in phenotypic responses of plant species has increased over the years. Approximately 90% of the studies were performed in the last decade. Studies that used DNA methylation to access epigenetic processes were more numerous than the other techniques throughout the studied period. The histone modification and miRNA techniques showed fluctuations in the number of publications over the last decade, with peaks alternating between these techniques (Figure 2a).

For abiotic factors, a pattern emerged across the years, with water as the factor with the greatest number of studies. Although it has a higher total number of studies than the water studies, temperature was the second most studied factor over the period, followed by salinity (Figure 2b).

### 2.4. Geographical Distribution of the Studies

According to our results, China is the country with the largest number of published studies (40%), followed by the USA, which accounted for only 7% of the total studies found. The first 10 countries with more studies are all from the Northern Hemisphere (Table 2). The first Southern Hemisphere country to appear in this list is Argentina (*n* = 8). Brazil is the 13th country in this ranking with only seven published studies.

### 2.5. Overlapping Techniques, Factors and Species

The three most commonly studied species are concentrated in 154 individual studies. This number represents 38% of the total number of selected studies (*n* = 401). Considering that 142 different species were studied and only the first five species have more than a dozen studies, the first three species have a significantly relevant value in this type of research. *A. thaliana* concentrates the greatest number of studies focused on histone modifications and temperature, while *O. sativa* has more studies on DNA methylation, divided between water and salinity. Finally, *Z. mays* had a higher number of studies on DNA methylation that included multiple abiotic factors (Figure 3).

## 3. Discussion

In this study, we aimed to investigate the main role of epigenetics in the response to abiotic stress in plants. The thorough reading and analysis of the 401 articles included in this review make it clear that epigenetic responses, due to not involving changes at the genomic level, are an effective strategy for stress response since they are both rapid and non-permanent. The plant under abiotic stress requires a quick response to avoid the individual’s death and consequently the inability to transmit their genes to the next generation.

### 3.1. The Influence of Epigenetic Responses to Abiotic Stress on Plants

We believe that the increase in epigenetic studies of abiotic factors reflects concern about human-induced global change [20]. Moreover, we cannot extrapolate this trend to biotic factors, such as herbivory or competition, due to the other physiological mechanisms involved in this process having not been analyzed [21]. This trend allows us to draw broader conclusions about epigenetic responses to abiotic factors as a whole. These responses are likely to be more extensive in plant species in the early stages of development, as observed in most of the studies included in this review. This shows how important this stage is for the establishment and survival of species.

The model of sympatric speciation published by John Maynard Smith in 1966 proposes that animal subpopulations acquire adaptations over time and make them reproductively isolated from other subpopulations of the same species. This process can be driven by factors such as preferences for different food resources, reproductive behaviors, or selective pressures [22]. However, for these changes to manifest at the genetic level, an extended period of evolution is required, which is typically marked by numerous hurdles and individual losses. When considering the necessity of short-term responses in plants, where adaptation must take place within a single generation, only epigenetic alterations can ensure such a phenotypic change that enables the individual to survive and breed even when challenged with abiotic stress. Therefore, the influence of epigenetic responses to abiotic stress seems to be related to the need for a rapid and efficient adaptation to environmental change that may be impermanent.

### 3.2. Most of the Studied Species Are of Economic Interest

The 10 most widely studied plant species are considered as biological model organisms with substantial economic value. Among these, *A. thaliana*, has been extensively researched, owing to its well-characterized genome. Notably, *A. thaliana* belongs to the same family as many important vegetable crops (Brassicaceae) including mustard, cabbage, and broccoli, which hold significant economic importance. Therefore, many genetic, botanical, and physiological studies conducted on *A. thaliana* can be extrapolated to these species [23].

Similarly, rice (*O. sativa*) and maize (*Z. mays*), which also have the most publications, are considered model species with great commercial value as they are the two of the largest food crops [24,25]. Recent research indicates that *Z. mays* has gradually gained importance as a domestic crop and will be the most widely traded variety in a few decades [26]. These prospects may indicate a new wave of research addressing the epigenetic responses of this species to abiotic stresses, so species of economic value remain among the most studied. Another advantage of using this model species with a well-established genetic database is the availability of extensive knowledge for researchers to use. Coupled with recent advances in genome sequencing, this has facilitated a wide range of studies using clones, hybrids, mutants, or specific genes and proteins [27]. Such studies have contributed to the dominant position of these plant species as the most widely investigated in the field of epigenetics, consistent with our own findings. In this way, the best understanding of epigenetics in non-model species against climate change should be obtained in a short time, considering the emergence of environment modification due to anthropic causes, which needs a fast phenotypic adjustment.

### 3.3. An Increase in Research in the Last Decade

Although biological and genetic research has been ongoing for a long time, the combination of epigenetic processes and plant responses to abiotic stressors has increased significantly in the last decade. Since then, epigenetics has explained how changes that occur in response to environmental changes can be transmitted without altering the DNA sequence [28,29]. In addition to the extensive genomic knowledge on model species, this kind of research has become crucial, especially to elucidate how epigenetic mechanisms impact the evolutionary process of species [30]. For this reason, in addition to studies on crop breeding, further research is needed to explain how changes that are not answered by genetic variation itself (epigenetics) can be inherited between generations. These responses will help in understanding the role of these mechanisms in adaptive processes in plant species and explain the reasons by which environmental factors affect plant phenotypes [11,30,31].

The disparity in production between the Northern and Southern Hemispheres underscores the disparity in financial investment in research [32]. Leading countries in publication rankings exhibit an interest in investing in scientific research for economic gains. Despite, the presence of exceptional researchers and research centers in the Southern Hemisphere, most research can not focus on biodiversity conservation due to economic constraints. In this sense substantial investments in research require decisions that may not prioritize other pressing needs of local populations, further complicating matters. This entire scenario raises concerns, particularly regarding regions rich in biodiversity. Nevertheless, a global comprehension of species responses in the Northern Hemisphere enables us to draw preliminary conclusions about the similarities and disparities in the responses of native species in the Southern Hemisphere. While further research is warranted to delineate these nuances accurately, it serves as an initial reference point for future investigations.

### 3.4. Leading Country of Publication

When examining the geographic location where studies were conducted based on the country of the first author or the location of the experiment (when it was possible to extract from the paper information), we found that there was a difference between the number of publications in the Northern and Southern Hemispheres. Most studies involving epigenetics to understand the species’ responses of stress from abiotic factors have been developed in the Northern Hemisphere. China and the United States are in the top of this ranking. However, the dominance of these countries was expected, as both are among the countries that allocate more domestic costs to research and development, according to the Organization for Economic Cooperation and Development. China has established itself as the second-best country for research in science and technology indicators since 2010. Moreover, due to its growth, China is gradually catching up with the U.S., while the other country remains stable [33].

Another feature that places China at the forefront of the most research-intensive countries is its strong government support [33] and, previously, the integration of science discovered in universities has been practiced in its economy. These characteristics have made China a powerhouse in scientific research. Our results demonstrate that economic interest serves as a driving force for innovation in scientific and technological research This is confirmed by the ten most studied species (Table 1), most of which have high economic value and whose discoveries enable scientific research improvements for them. By these means, in addition to new discoveries, opportunities for improvement are created, reflecting the increasing rate of these species production in several aspects, incorporating innovative development and new research results into practice [33]. Therefore, the main effect of the geographical distribution of research is related to the species studied. And the most studied abiotic factors (water and temperature) reflect agronomic interests, such as water scarcity having a negative effect on a crop [34,35]. However, in the current scenario, these two factors are becoming increasingly important due to global warming and the extreme climatic changes that the planet is experiencing. Therefore, in areas of biodiversity where there are few studies on the epigenetic responses of plants, these studies will be a reference on how to reproduce them in these ecosystems.

On the other hand, we note a gap in studies conducted in the Southern Hemisphere, a region with tropical forests that have the greatest biodiversity in the world, especially in plant species [36,37]. This is true for the Amazon and Atlantic forests in Brazil, which have a large number of species with high levels of endemism and the highest risk and threat due to anthropogenic activities that result in deforestation [37,38,39]. Investment in science and the expertise developed over the years in the Northern Hemisphere can help to develop these aspects in these regions. For this reason, scientific partnerships between the US and China with countries in the Southern Hemisphere that have high biodiversity would be an excellent way to reduce the knowledge gaps in this type of research and thus increase the number of studies in these tropical regions. Therefore, an alternative to reducing the scientific gap found in these studies in relation to these countries is to increase investments and partnerships between northern and southern countries. Therefore, future research should prioritize these regions as they host many endangered species. Tropical environments could take advantage of studies investigating the regulation of abiotic factors, forming the basis for defining genetic conservation and strategic cultivation.

### 3.5. DNA Methylation Is the Most Studied Epigenetic

Epigenetic changes refer to modifications that do not alter nucleotide sequences themselves. Instead, they alter the availability of DNA for protein binding, thereby inducing or suppressing gene transcripts [3]. In this study, we found that the mechanism of DNA methylation is the most frequently studied. DNA methylation studies are widespread because there is already a body of knowledge, protocols and applications that enable the use of this technique [40]. For example, it is possible to identify patterns of histone modifications using DM studies. There is also evidence that methylation changes can be passed on from one generation to the next [41,42]. We believe that there will be a tendency to perform studies using different integrated techniques, as for each technique there is already a considerable database (mainly for model species) that provides the epigenetic responses of plants to abiotic factors [20,43]. This result was expected since DNA methylation has been the most studied technique in epigenetic research for some time [44,45]. Several studies have demonstrated the importance of this modification in maintaining the integrity of the genome by repressing or activating DNA transcription [46]. Therefore, the potential of these mechanisms remains appreciated as they contribute to understanding and predicting changes in plants exposed to environmental changes. 

However, although they are widely used, they can also be improved when studies are conducted with other epigenetic mechanisms. We have observed that incorporating this mechanism alongside others, such as histone modifications, contributes to the comprehension of how plants respond to environmental factors. It also helps to clarify which mechanisms are involved in these adaptations. It is important to highlight that recent research has also aimed to comprehend how one mechanism can impact or modify another process. It should be emphasized that there may be a gap in how epigenetic responses occur when considering the synergy of multiple techniques simultaneously. This unveils a new pathway for studying the interactions of these processes, which can no longer be studied in isolation, as was the case a few decades ago.

Another point that stood out in our analysis was the large number of research papers dealing with clones, hybrids or transgenics of the main species of economic value. This is due to plants with such traits, when exposed to stressors, showing changes in DNA methylation. Thus, we can conclude that many of these studies using species of economic value were conducted in countries such as China, which have greater economic potential.

### 3.6. Temperature and Water Proportionately Studied

Plants are often exposed to unpredictable environmental changes, a factor that is becoming increasingly common due to climate change. Therefore, these sessile organisms are exposed to various environmental stresses. In the context of climate change, the most common stressors are caused by changes in temperature, water availability, light, and nutrients [16,40]. Epigenetic dynamics are critical for genetic regulation in the face of these stressors.

The distribution of abiotic stressors reflects a more common and likely scenario in different ecosystems. Salt stress is the least common of all and occurs most frequently in cultivated areas [47]. Light stress usually occurs during the opening of clearings caused by natural factors such as a storm, or during a change of habitat when a species is removed from a greenhouse into an open environment [48]. Heat stress has become more common due to global warming [32]. Studies have shown that even small increases in temperature can seriously damage plant species [32,49]. Finally, water stress, which is associated with extreme climatic events such as prolonged droughts or floods, damages plant species and often severely affects agricultural production [34,35].

Increased research on temperature, in our case in relation to low temperatures, has been largely enabled by existing knowledge of genes regulating this factor. For example, DNA methylation and histone modifications are already known to regulate certain temperature-dependent genes [50]. These mechanisms act at the transcriptional level by altering the chromatin status of the gene of interest. Regarding water, we found a much larger number of studies on drought and water deficit, which is consistent with what is also found in the current literature. However, it is worth noting that the majority of studies are focused on understanding the effects of this stress during a crop stage, such as twining or seeding. Again, prior knowledge of the genes that regulate this stress has allowed for a different response at different stages.

In addition, knowledge of the phytohormone ABA is an important resource for drought resistance because it is synthesized in plants in response to drought stress. The mechanism of this hormone is a starting point for understanding the epigenetic mechanisms underlying responses to drought stress, along with possible changes in the ABA synthesis pathway [51,52,53]. Hence, it is critical that future research endeavors focus on elucidating these processes in other key plant developmental stages, including seedlings, saplings, and adults, as environmental factors can significantly impact all of these stages.

### 3.7. Research Deficits in the Tropics and in Native Species

As mentioned above, the selected studies focused primarily on commercial species due to their extensive genetic database and economic value. In addition, the use of genetic varieties or clones allowed for a clearer isolation of environmental or genetic effects. For non-commercial species, breeding studies could provide valuable insights into less studied species. This allows searching for specific groups of specimens for defined purposes. For instance, selecting individuals with a higher concentration of ABA to introduce them into ecological plantations.

However, it is remarkable how little demand exists for studies involving tropical native species in the Southern Hemisphere, especially for forest species. This hemisphere hosts some biodiversity hotspots such as the Atlantic Forest [38], Madagascar and Indian Ocean Islands, and East Melanesian Islands. Despite the advantages of using economic species, we believe that more studies should be conducted focusing on ecological interests. This is particularly urgent since anthropogenic impacts have serious effects on the environment and impair the environmental services provided by forests [37,38,39] (Wright 2010). In the context of climate change, there are extreme climate events for which we have no information on how plant organisms are affected at the epigenetic level (Wright 2010) [54,55]. By those means, there is currently little to no information on how threatened species will respond to these extreme events. The absence of tropical species in the studies selected here highlights a knowledge gap in epigenetic research related to abiotic stressors for tropical plants. This lack of knowledge is a concern as we need to understand how these species will persist in the face of climate change.

It is well known that environmental changes are one of the most important genetic selectors [56,57,58], and epigenetics can influence this selection when the mechanisms are heritable but do not involve genetic changes [30,59]. Thus, understanding the processes and effects of these epigenetic mechanisms and their heritability allows us to propose environmental strategies aimed at the conservation of biodiversity. Finally, with this work we realize that it is possible to extend epigenetics techniques to investigate new areas and species to achieve more knowledge in the adaptation field. However, a deeper examination of non-commercial species, which are equally crucial for environmental health and biodiversity, is required.

## 4. Materials and Methods

### 4.1. Search and Database

For this review, we compiled articles published through July 2022 to analyze the role of epigenetics in the response of plant species to abiotic stressors. We searched by title, abstract, and keywords in the Web of Science and Scopus databases. Thus, we had a set of terms based on epigenetics, plant, and phenotypic plasticity, as shown in Box 1, which we applied to both databases. We applied filters to exclude reviews and include only articles. After that, we applied a second filter to find only articles from the following fields: Biochemistry, Genetics and Molecular Biology, Environmental and Forest Sciences, and Agricultural and Life Sciences. We found 6143 articles matching these filters. We used the “bibliometrix” package of the R 4.2.1 program to eliminate the duplicate articles, obtaining 5772 final studies.


Box 1Set of terms based on epigenetics, plants and phenotypic plasticity used to search. the databases.(epigenetic* OR “DNA methylation” OR “epigenetic inheritance” OR “transgenerational epigenetic” OR “epigenetic regulation*” OR “ecological epigenetic*” OR “heritable epigenetic” OR “epigenetic change*” OR “epigenetic variation” OR “histone modification” OR “MicroRNA”) AND (plant* OR tree*) AND (phenotypic* OR “phenotypic plasticity” OR “environmental change*” OR “morphological variation*” OR “environmental stress*”)) 

In selecting articles to include in the analysis, we applied certain criteria. By reading the title and abstract, we excluded all research that did not focus exclusively on plant species. We also excluded articles that were not in English or that were book chapters or gray literature. Next, we selected articles that simultaneously addressed abiotic factors and epigenetics. We did not include studies dealing with vernalization or cryopreservation because they did not fit our objectives. Thus, we followed the established protocol, which includes the following steps: (i) identification, (ii) screening, (iii) eligibility, and (iv) inclusion (Figure 4). Finally, the selected articles were read in full and those that did not meet the inclusion criteria were excluded. A total of 401 articles were considered for information extraction.

### 4.2. Categorization and Data Analysis

After completing the database, we extracted from all articles (1) the name of the species, (2) the type of abiotic stress studied, (3) the type of epigenetic technique studied, (5) the geographic coordinates. To determine the pattern in the number of publications through 2022, the temporal trend of published articles was also analyzed using the statistical program R 3.5.2 (R Core Team 2020).

## 5. Conclusions

Our review leads us to conclude that epigenetic studies on plant species exposed to abiotic factors are extremely important to understand plant establishment and survival, whether in the face of factors of agronomic and/or ecological interest (such as climate change). Although the most studied factors and the most commonly used techniques are geographically diverse, it can be concluded that these studies provide a fundamental basis for application to non-model species and native species in regions of high biodiversity. As research with model species and the rise of epigenetic studies in recent decades allows us, through taxonomic proximity, we need to mitigate trial and error and move to more robust and assertive research when it comes to the ecosystem we are interested in. Finally, the knowledge gap in native and non-economic species can be reduced by mitigating crucial factors that are independent of researchers, such as economic support and valorization of scientific discoveries.

## Figures and Tables

**Figure 1 plants-13-02082-f001:**
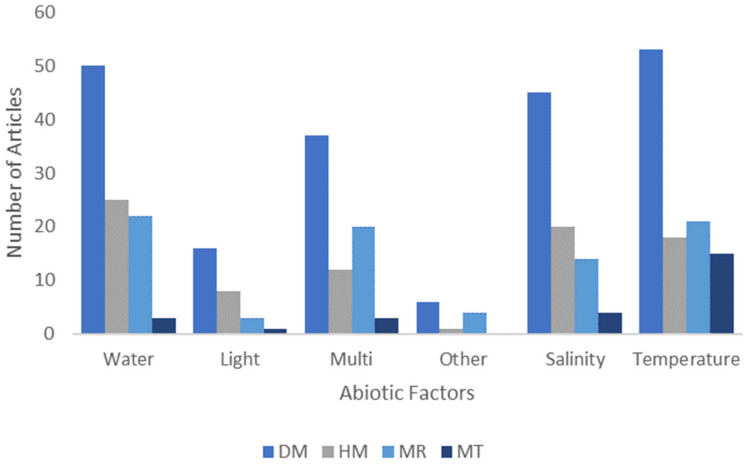
Number of published articles for each abiotic factor and epigenetic techniques used in each case. DM (DNA methylation), HM (modified histones), MR (microRNA studies), and MT (multiple techniques).

**Figure 2 plants-13-02082-f002:**
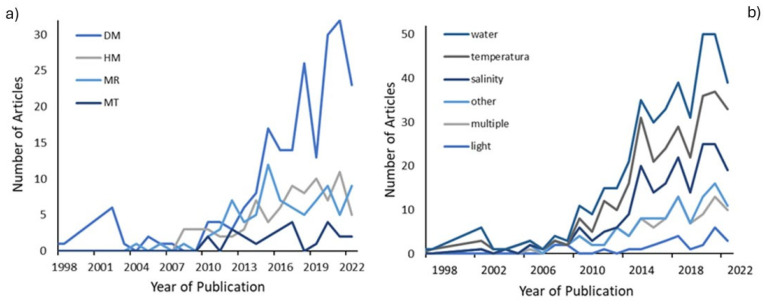
Temporal distribution of studies. (**a**) Distribution of epigenetic techniques: DM (DNA methylation), HM (modified histones), MR (studies with microRNAs), and MT (multiple techniques). (**b**) Distribution of abiotic factor groups.

**Figure 3 plants-13-02082-f003:**
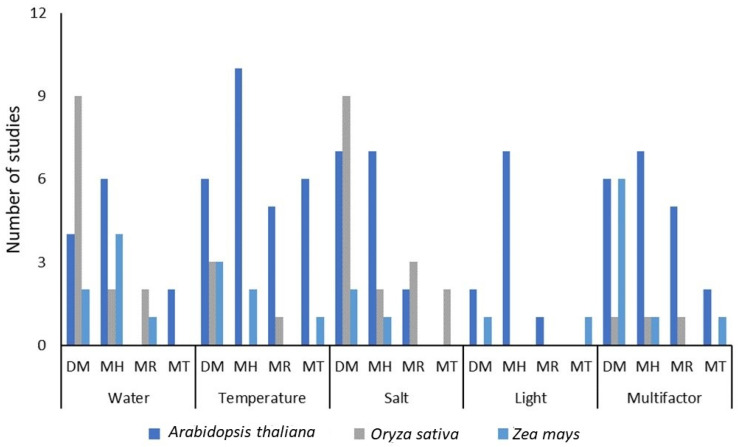
Comparative diagram of epigenetic mechanisms and major abiotic factors for the three most studied species. DM (DNA methylation), MH (modified histones), MR (studies with microRNAs), and MT (multiple techniques).

**Figure 4 plants-13-02082-f004:**
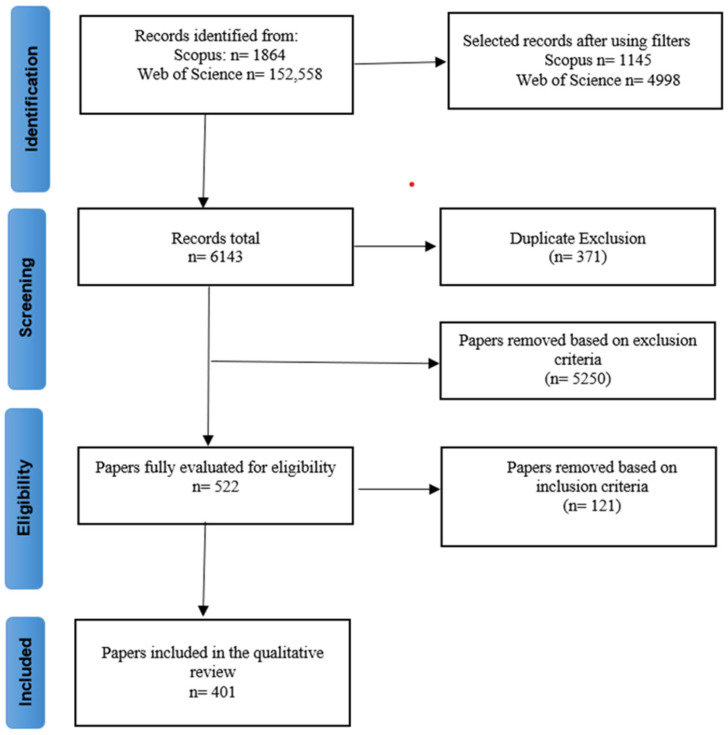
Steps for selecting studies for the review. The numbers in parentheses indicate the number of studies excluded.

**Table 1 plants-13-02082-t001:** Number of studies by species related to abiotic stress and epigenetics published from 1997 to 2022 and indexed in Web of Science and Scopus.

Species	Family	Number of Studies
*Arabidopsis thaliana*	Brassicaceae	90
*Oryza sativa*	Poaceae	38
*Zea mays*	Poaceae	26
*Hordeum vulgare*	Poaceae	11
*Triticum aestivum*	Poaceae	11
*Glycine max*	Leguminosae	9
*Gossypium hirsutum*	Malvaceae	8
*Brassica napus*	Brassicaceae	7
*Nicotiana tabacum*	Solanaceae	6
*Solanum lycopersicum*	Solanaceae	6

**Table 2 plants-13-02082-t002:** Number of studies per country on epigenetics and abiotic factors, from 1997 to 2022.

Country	Number of Articles
China	158
USA	30
India	21
Germany	17
Japan	16
Italy	15
Spain	14
Poland	10
Canada	9
France	9

## Data Availability

The analyzed database in this current study may be available from the corresponding author on reasonable request.

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
