# Peer review of "Epigenetics Regulation in Responses to Abiotic Factors in Plant Species: A Systematic Review"

_plants, 2024, doi:10.3390/plants13152082_

Round 1

Reviewer 1 Report

Comments and Suggestions for Authors

The manuscript provides a systematic review of the literature on the role of epigenetics in the response of plant species to abiotic stressors. The study analysed a total of 401 articles and identified the most studied plant species, the type of abiotic factors studied, and the epigenetic techniques most used in the scientific literature. The study also highlighted the gap in research on non-economic and tropical plant species, emphasising the need for further research in these areas. The methods section of the document outlines the search and database compilation process, the criteria for selecting articles, and the steps for categorisation and data analysis.

The article is enriched with several figures that are instrumental in comprehending the data and analysis presented. Notably, Figure 1 visually guides the selection process of studies from the review, while Figure 2 vividly illustrates the temporal trend of published articles. Furthermore, Figure 3 provides a comprehensive distribution of studies across different species and epigenetic techniques, offering valuable insights into the research landscape. These figures serve as a visual aid, significantly enhancing the understanding of the data and findings presented in the article.

Overall, this review is a valuable resource for researchers and practitioners interested in the intersection of epigenetics and plant adaptation to environmental stressors. And may act as a guideline for future research themes.

Author Response

We acknowledge the reviewers' suggestions and comments

Reviewer 1:

Comments and Suggestions for Authors

The manuscript provides a systematic review of the literature on the role of epigenetics in the response of plant species to abiotic stressors. The study analysed a total of 401 articles and identified the most studied plant species, the type of abiotic factors studied, and the epigenetic techniques most used in the scientific literature. The study also highlighted the gap in research on non-economic and tropical plant species, emphasising the need for further research in these areas. The methods section of the document outlines the search and database compilation process, the criteria for selecting articles, and the steps for categorisation and data analysis.

The article is enriched with several figures that are instrumental in comprehending the data and analysis presented. Notably, Figure 1 visually guides the selection process of studies from the review, while Figure 2 vividly illustrates the temporal trend of published articles. Furthermore, Figure 3 provides a comprehensive distribution of studies across different species and epigenetic techniques, offering valuable insights into the research landscape. These figures serve as a visual aid, significantly enhancing the understanding of the data and findings presented in the article.

Overall, this review is a valuable resource for researchers and practitioners interested in the intersection of epigenetics and plant adaptation to environmental stressors. And may act as a guideline for future research themes.

R: Thank you very much for your comments. We are glad to read them. Your words encourage us to continue our scientific research.

Reviewer 2 Report

Comments and Suggestions for Authors

The goal of this review is to examine the temporal and geographic aspects 98 of epigenetic research related to abiotic stressors.

The manuscript is mainly the summary of the particular research field. It would be good if authors can justify or provide rationale for the obtained results.

This MS needs major adjustments;

Title is misleading and does not corresponds to the abstract. Phenotypic response against which kind of factor? In abstract author has mentioned abiotic factors only. Be specific in title also.

Line 17; author has mentioned that they have performed systematic review of literature already available for example up to the year 2022. What about the study that has been conducted last two years [2023-2024]? for example;

Abdulraheem, M.I.; Xiong, Y.; Moshood, A.Y.; Cadenas-Pliego, G.; Zhang, H.; Hu, J. Mechanisms of Plant Epigenetic Regulation in Response to Plant Stress: Recent Discoveries and Implications. Plants 202413, 163. https://doi.org/10.3390/plants13020163.

Line 148; The time that is shown in Figure 2a&b; is only to the year 2021; it would be better can add the data from the year 2023 & 2024 as well.

Line 165; The study highlights China as the country with the largest number of published studies on epigenetics in plants. How does the geographical distribution of research impact the diversity of methodologies, species studied, and abiotic factors investigated? Are there any research gaps that could influence the global understanding of epigenetic responses in plants?

Line 154; 300; The distribution of studies across abiotic factors shows a strong focus on temperature and water availability, with fewer studies on salinity and light factors. How does this distribution align with the actual environmental stresses faced by plants, particularly in different ecosystems? Authors are requested to provide the rationale of their findings. In addition; The study notes a significant increase in publications on epigenetic regulation over the last decade, with DNA methylation studies being the most numerous. How does this temporal trend reflect the evolving understanding of epigenetics in plant responses to abiotic stress? Provide the rationale for these observation as well.

Line 210; The study notes an increase in epigenetics research related to abiotic stress in the last decade. However, does this trend reflect a comprehensive understanding of epigenetic responses across various stressors and developmental stages? How might the evolving research focus influence future directions in studying epigenetic mechanisms in non-model or native species?

Line 247; It would be better if author can discuss the potential consequences of the geographical disparity in epigenetics research, especially regarding the lack of studies in biodiversity-rich regions like the Southern Hemisphere? How might this imbalance impact our overall comprehension of plant adaptations to environmental challenges?

Line 259; Considering the influence of economic factors on research intensity, what strategies can be implemented to encourage more diverse and inclusive research agendas that encompass non-commercial species and regions with high biodiversity? How can international collaborations contribute to addressing these research gaps?

Line 280; While DNA methylation is extensively studied, how effectively are other epigenetic mechanisms, such as histone modifications, integrated into current research? Are there specific challenges or gaps in understanding the synergistic effects of different epigenetic processes in plant responses to environmental stressors?

I did not find any conclusion section.

Author Response

Reviewer 2:

Comments and Suggestions for Authors

The goal of this review is to examine the temporal and geographic aspects 98 of epigenetic research related to abiotic stressors.

The manuscript is mainly the summary of the particular research field. It would be good if authors can justify or provide rationale for the obtained results.

This MS needs major adjustments;

R: We totally agree with the reviewer and improve the text in order to attend the suggestion.

Title is misleading and does not corresponds to the abstract. Phenotypic response against which kind of factor? In abstract author has mentioned abiotic factors only. Be specific in title also.

R: We all agree. So the new title is: Epigenetic regulation in response to abiotic factors in plant species: a systematic review

Line 17; author has mentioned that they have performed systematic review of literature already available for example up to the year 2022. What about the study that has been conducted last two years [2023-2024]? for example;

Abdulraheem, M.I.; Xiong, Y.; Moshood, A.Y.; Cadenas-Pliego, G.; Zhang, H.; Hu, J. Mechanisms of Plant Epigenetic Regulation in Response to Plant Stress: Recent Discoveries and Implications. Plants 202413, 163. https://doi.org/10.3390/plants13020163.

R: We are sorry, but we could not attend this suggestion at this short time. Although we think it would be interesting to add articles from 2023 and 2024 (except for reviews that meet our inclusion and exclusion criteria), we believe that our review has a solid database that has allowed us to consolidate the results presented. We strongly agreed that the addition of these two years would not change the scenarios found, although it would make the review more robust in terms of data. Also, the main reason for not follow your suggestion for updating is the 10-day period for reviewers to return comments. Unfortunately, this is a short time for this process, since we should start all the analysis from the beginning.

Line 148; The time that is shown in Figure 2a&b; is only to the year 2021; it would be better can add the data from the year 2023 & 2024 as well.

R: Thank you for your suggestion, but the issue with the graph is the scale. The data goes beyond the point that refers to the year 2021 since our data goes to 2022. Unfortunately, changing this scale messes up the whole layout of the figure.

Line 165; The study highlights China as the country with the largest number of published studies on epigenetics in plants. How does the geographical distribution of research impact the diversity of methodologies, species studied, and abiotic factors investigated? Are there any research gaps that could influence the global understanding of epigenetic responses in plants?

R: Thank you for your comments. We include a text in lines 288-295 to clarify and attend to your suggestion.

Line 154; 300; The distribution of studies across abiotic factors shows a strong focus on temperature and water availability, with fewer studies on salinity and light factors. How does this distribution align with the actual environmental stresses faced by plants, particularly in different ecosystems? Authors are requested to provide the rationale of their findings. In addition; The study notes a significant increase in publications on epigenetic regulation over the last decade, with DNA methylation studies being the most numerous. How does this temporal trend reflect the evolving understanding of epigenetics in plant responses to abiotic stress? Provide the rationale for these observation as well.

R: We have rewritten this part of the discussion (lines  350-358) to improve the text as recommended.

Line 210; The study notes an increase in epigenetics research related to abiotic stress in the last decade. However, does this trend reflect a comprehensive understanding of epigenetic responses across various stressors and developmental stages? How might the evolving research focus influence future directions in studying epigenetic mechanisms in non-model or native species?

R:  Thank you for your suggestion. We have improved the text in lines  197-204 as recommended.

Line 247; It would be better if author can discuss the potential consequences of the geographical disparity in epigenetics research, especially regarding the lack of studies in biodiversity-rich regions like the Southern Hemisphere? How might this imbalance impact our overall comprehension of plant adaptations to environmental challenges?”

R: We have rewritten the text (lines 253-265) to better clarify the topic, as recommended.

Line 259; Considering the influence of economic factors on research intensity, what strategies can be implemented to encourage more diverse and inclusive research agendas that encompass non-commercial species and regions with high biodiversity? How can international collaborations contribute to addressing these research gaps?

R: We agree with the suggestion and have added lines  300-307 to comply with the suggestion.

Line 280; While DNA methylation is extensively studied, how effectively are other epigenetic mechanisms, such as histone modifications, integrated into current research? Are there specific challenges or gaps in understanding the synergistic effects of different epigenetic processes in plant responses to environmental stressors?

R: We have improved the discussion by adding text to lines 315- 322 and 333-335 to clarify the issue as suggested. 

I did not find any conclusion section.

R: We fully agree that there is no conclusion. We have added it in lines 444-455.

Round 2

Reviewer 2 Report

Comments and Suggestions for Authors

The author has incorporated all the suggestions.